# Geometric Dataset Distances via Optimal Transport

**David Alvarez-Melis**
Microsoft Research, New England
alvarez.melis@microsoft.com

**Nicolò Fusi**
Microsoft Research, New England
nfusi@microsoft.com

## Abstract

The notion of task similarity is at the core of various machine learning paradigms, such as domain adaptation and meta-learning. Current methods to quantify it are often heuristic, make strong assumptions on the label sets across the tasks, and many are architecture-dependent, relying on task-specific optimal parameters (*e. g.,* require training a model on each dataset). In this work we propose an alternative notion of distance between datasets that (i) is model-agnostic, (ii) does not involve training, (iii) can compare datasets even if their label sets are completely disjoint and (iv) has solid theoretical footing. This distance relies on optimal transport, which provides it with rich geometry awareness, interpretable correspondences and well-understood properties. Our results show that this novel distance provides meaningful comparison of datasets, and correlates well with transfer learning hardness across various experimental settings and datasets.

## 1 Introduction

A key hallmark of machine learning practice is that labeled data from the application of interest is usually scarce. For this reason, there is vast interest in methods that can combine, adapt and transfer knowledge across datasets and domains. Entire research areas are devoted to these goals, such as domain adaptation, transfer-learning and meta-learning. A fundamental concept underlying all these paradigms is the notion of *distance* (or more generally, *similarity*) between datasets. For instance, transferring knowledge across similar domains should intuitively be easier than across distant ones. Likewise, given a choice of various datasets to pretrain a model on, it would seem natural to choose the one that is closest to the task of interest.

Despite its evident usefulness and apparent simpleness, the notion of distance between datasets is an elusive one, and quantifying it efficiently and in a principled manner remains largely an open problem. Doing so requires solving various challenges that commonly arise precisely in the settings for which this notion would be most useful, such as the ones mentioned above. For example, in supervised machine learning settings the datasets consist of both features and labels, and while defining a distance between the former is often —though not always— trivial, doing so for the labels is far from it, particularly if the label-sets across the two tasks are not identical (as is often the case for off-the-shelf pretrained models).

Current approaches to transfer learning that seek to quantify dataset similarity circumvent these challenges in various ingenious, albeit often heuristic, ways. A common approach is to compare the dataset via proxies, such as the learning curves of a pre-specified model [37] or its optimal parameters [2, 32] on a given task, or by making strong assumptions on the similarity or co-occurrence of labels across the two datasets [50]. Most of these approaches lack guarantees, are highly dependent on the probe model used, and require training a model to completion (*e. g.,* to find optimal parameters) on each dataset being compared. On the opposite side of the spectrum are principled notions of discrepancy between domains [9, 41], which nevertheless are often not computable in practice, or do not scale to the type of datasets used in machine learning practice.

In this work, we seek to address some of these limitations by proposing an alternative notion of distance between datasets. At the heart of this approach is the use of optimal transport (OT) distances [52] to compare distributions over feature-label pairs in a geometrically-meaningful and principled way. In particular, we propose a hybrid Euclidean-Wasserstein distance between feature-label pairs across domains, where labels themselves are modeled as distributions over features vectors. As a consequence of this technique, our framework allows for comparison of datasets *even if their label sets are completely unrelated or disjoint*, as long as a distance between their features can be defined. This notion of distance between labels, a by-product of our approach, has itself various potential uses, *e. g.,* to optimally sub-sample classes from large datasets for more efficient pretraining.

In summary, we make the following contributions:
- We introduce a principled, flexible and efficiently computable notion of distance between datasets
- We propose algorithmic strategies to scale up computation of this distance to very large datasets
- We provide extensive empirical evidence that this distance is highly predictive of transfer learning success across various domains, tasks and data modalities

## 2   Related Work

**Discrepancy Distance**    Various notions of (dis)similarity between data distributions have been proposed in the context of domain adaptation, such as the $d_A$ [9] and discrepancy distances[1] [41]. These discrepancies depend on a loss function and hypothesis (*i. e.,* predictor) class, and quantify dissimilarity through a supremum over this function class. The latter discrepancy in particular has proven remarkably useful for proving generalization bounds for adaptation [13], and while it can be estimated from samples, bounding the approximation quality relies on quantities like the VC-dimension of the hypothesis class, which might not be always known or easy to compute.

**Dataset Distance via Parameter Sensitivity**    The Fisher information metric is a classic notion from information geometry [6, 8] that characterizes a parametrized probability distribution locally through the sensitivity of its density to changes in the parameters. In machine learning, it has been used to analyze and improve optimization approaches [7] and to measure the capacity of neural networks [39]. In recent work, Achille et al. [2] use this notion to construct vector representations of tasks, which they then use to define a notion of similarity between these. They show that this notion recovers taxonomic similarities and is useful in meta-learning to predict whether a certain feature extractor will perform well in a new task. While this notion shares with ours its agnosticism of the number of classes and their semantics, it differs in the fact that it relies on a probe network trained on a specific dataset, so its geometry is heavily influenced by the characteristics of this network. Besides the Fisher information, a related information-theoretic notion of complexity that can be used to characterize tasks is the Kolmogorov Structure Function [38], which Achille et al. [1] use to define a notion of *reachability* between tasks.

**Optimal Transport-based distributional distances**    The general idea of representing complex objects via distributions, which are then compared through optimal transport distances, is an active area of research. Also driven by the appeal of their closed-form Wasserstein distance, Muzellec and Cuturi [44] propose to embed objects as elliptical distributions, which requires differentiating through these distances, and discuss various approximations to scale up these computations. Frogner et al. [25] extend this idea but represent the embeddings as discrete measures (*i. e.,* point clouds) rather than Gaussian/Elliptical distributions. Both of these works focus on embedding and consider only within-dataset comparisons. Also within this line of work, Delon and Desolneux [19] introduce a Wasserstein-type distance between Gaussian mixture models. Their approach restricts the admissible transportation couplings themselves to be Gaussian mixture models, and does not directly model label-to-label similarity. More generally, the Gromov-Wasserstein distance [43] has been proposed to compare collections across different domains [4, 42], albeit leveraging only features, not labels.

**Hierarchical OT distances**    The distance we propose can be understood as a hierarchical OT distance, *i. e.,* one where the ground metric itself is defined through an OT problem. This principle has been explored in other contexts before. For example, Yurochkin et al. [55] use a hierarchical OT distance for document similarity, defining a inner-level distance between topics and a outer-level distance between documents using OT. Dukler et al. [22] on the other hand use a nested Wasserstein distance as a loss for generative model training, motivated by the observation that the Wasserstein

distance is better suited to comparing images than the usual pixel-wise $L_2$ metric used as ground metric. Both the goal and the actual metric used by these approaches differs from ours.

**Optimal Transport with Labels**   Using label information to guide the optimal transport problem towards class-coherent matches has been explored before, *e. g.,* by enforcing group-norm penalties [15] or through submodular cost functions [5]. These works are focused on the unsupervised domain adaptation setting, so their proposed modifications to the OT objective use only label information from one of the two domains, and even then, do so without explicitly defining a metric between these. Furthermore, they do not lead to proper distances, and these works deal with a single static pair of tasks, so they lack analysis of the distance across multiple source and target datasets. Closest to this work is JDOT [14] and its extension by Damodaran et al. [17], which use a hybrid feature-label transportation cost that quantifies discrepancy of labels through a classification loss (e.g., hinge-loss). As a consequence, this approach requires the two distributions share the exact same label set, might not yield a true metric depending on the loss chosen, and requires careful scaling of the two components of the cost.

## 3   Background on Optimal Transport

Optimal transport (OT) is a powerful and principled approach to compare probability distributions with deep theoretical foundations [52, 53] and desirable computational properties [46]. It considers a complete and separable metric space $\mathcal{X}$, along with probability measures $\alpha \in \mathcal{P}(\mathcal{X})$ and $\beta \in \mathcal{P}(\mathcal{X})$. These can be continuous or discrete measures, the latter often used in practice as empirical approximations of the former whenever working in the finite-sample regime.  The Kantorovich formulation [31] of the transportation problem reads:

$$\mathrm{OT}(\alpha, \beta) \triangleq \min_{\pi \in \Pi(\alpha, \beta)} \int_{\mathcal{X} \times \mathcal{X}} c(x, y)\, \mathrm{d}\pi(x, y), \tag{1}$$

where $c(\cdot, \cdot) : \mathcal{X} \times \mathcal{X} \to \mathbb{R}^+$ is a cost function (the "ground" cost), and the set of couplings $\Pi(\alpha, \beta)$ consists of joint distributions over the product space $\mathcal{X} \times \mathcal{X}$ with marginals $\alpha$ and $\beta$:

$$\Pi(\alpha, \beta) \triangleq \{\pi \in \mathcal{P}(\mathcal{X} \times \mathcal{X}) \mid P_{1\#}\pi = \alpha, P_{2\#}\pi = \beta\}. \tag{2}$$

Whenever $\mathcal{X}$ is equipped with a metric $d_{\mathcal{X}}$, it is natural to use it as ground cost, *e. g.,* $c(x, y) = d_{\mathcal{X}}(x, y)^p$ for some $p \geq 1$. In such case, $\mathrm{W}_p(\alpha, \beta) \triangleq \mathrm{OT}(\alpha, \beta)^{1/p}$ is called the $p$-Wasserstein distance. The case $p = 1$ is also known as the Earth Mover's Distance [48].

The measures $\alpha$ and $\beta$ are rarely known in practice. Instead, one has access to finite samples $\{\mathbf{x}^{(i)}\} \in \mathcal{X}, \{\mathbf{y}^{(j)}\} \in \mathcal{X}$, which implicitly define discrete measures $\alpha = \sum_{i=1}^{n} \mathbf{a}_i \delta_{\mathbf{x}^{(i)}}$ and $\beta = \sum_{i=1}^{m} \mathbf{b}_i \delta_{\mathbf{y}^{(j)}}$, where $\mathbf{a}, \mathbf{b}$ are vectors in the probability simplex, and the pairwise costs can be compactly represented as an $n \times m$ matrix $\mathbf{C}$, *i. e.,* $\mathbf{C}_{ij} = c(\mathbf{x}^{(i)}, \mathbf{y}^{(j)})$. In this case, Eq. (1) becomes a linear program, whose cubic complexity is often prohibitive. The entropy-regularized problem

$$\mathrm{OT}_\epsilon(\alpha, \beta) \triangleq \min_{\pi \in \Pi(\alpha, \beta)} \int_{\mathcal{X} \times \mathcal{X}} c(x, y)\, \mathrm{d}\pi(x, y) + \varepsilon \mathrm{H}(\pi \,|\, \alpha \otimes \beta), \tag{3}$$

where $\mathrm{H}(\pi \mid \alpha \otimes \beta) = \int \log(\mathrm{d}\pi/\mathrm{d}\alpha\, \mathrm{d}\beta)\, \mathrm{d}\pi$ is the relative entropy, can be solved much more efficiently —and with better sample complexity [28]— by using the Sinkhorn algorithm [3, 16], which enables a time/accuracy trade-off through $\varepsilon$. The *Sinkhorn divergence* [27], defined as

$$\mathrm{SD}_\varepsilon(\alpha, \beta) = \mathrm{OT}_\varepsilon(\alpha, \beta) - \frac{1}{2}\mathrm{OT}_\varepsilon(\alpha, \alpha) - \frac{1}{2}\mathrm{OT}_\varepsilon(\beta, \beta), \tag{4}$$

has many useful properties: it is positive, convex, and metrizes convergence of measures [24].

## 4   Optimal Transport between Datasets

The definition of *dataset* is notoriously inconsistent across the machine learning literature, sometimes referring only to features, or both features and labels. Here we are interested in supervised learning, so we define a dataset $\mathcal{D}$ as a set of feature-label pairs $(x, y) \in \mathcal{X} \times \mathcal{Y}$ over a certain feature space $\mathcal{X}$ and label set $\mathcal{Y}$. We will use the shorthand notations $z \triangleq (x, y)$ and $\mathcal{Z} \triangleq \mathcal{X} \times \mathcal{Y}$. Henceforth, we focus on the case of classification, so $\mathcal{Y}$ shall be a finite set. We consider two datasets $\mathcal{D}_A$ and $\mathcal{D}_B$, and assume, for simplicity, that their feature spaces have the same dimensionality, but

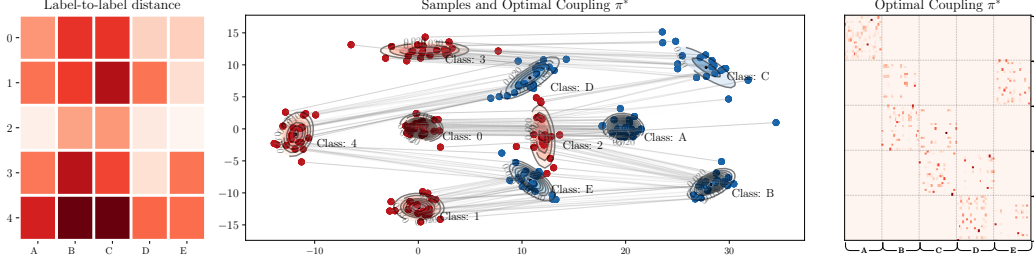

**Figure 1:** Our approach represents labels as distributions over features and computes Wasserstein distances between them (left). Combined with the usual metric between features, this yields a transportation cost between datasets. The optimal transport problem then characterizes the distance between them as the lowest possible cost to couple them (optimal coupling $\pi^*$ shown on the right).

will discuss how to relax this assumption later on. On the other hand, we make no assumptions on the label sets $\mathcal{Y}_A$ and $\mathcal{Y}_B$ whatsoever. In particular, the classes these encode could be partially overlapping or related (*e. g.,* ImageNet and CIFAR-10) or completely disjoint (*e. g.,* CIFAR-10 and MNIST). Although not a formal assumption of our approach, it will be useful to think of the samples in these two datasets as being drawn from joint distributions $P_A(x, y)$ and $P_B(x, y)$, *i. e.,* $\mathcal{D}_A = \{(x_A^{(i)}, y_A^{(i)})\}_{i=1}^n \sim P_A(x, y)$ and $\mathcal{D}_B = \{(x_B^{(j)}, y_B^{(j)})\}_{j=1}^m \sim P_B(x, y)$.

Our goal is to define a distance $d(\mathcal{D}_A, \mathcal{D}_B)$ without relying on external models or parameters. The interpretation above, viewed in light of Section 3, suggests comparing these datasets by computing an OT distance between their joint distributions. However, casting problem (1) in this context requires a —crucial— component: a metric on $\mathcal{Z}$, *i. e.,* between pairs $(x, y), (x', y')$. If we had metrics on $\mathcal{X}$ and $\mathcal{Y}$, we could define a metric on $\mathcal{Z}$ as $d_{\mathcal{Z}}(z, z') = \big(d_{\mathcal{X}}(x, x')^p + d_{\mathcal{Y}}(y, y')^p\big)^{1/p}$, for $p \geq 1$. In most applications, $d_{\mathcal{X}}$ is readily available, *e. g.,* as the euclidean distance in the feature space. On the other hand, $d_{\mathcal{Y}}$ will rarely be so, particularly between labels from unrelated label sets (*e. g.,* between cars in one image domain and and dogs in the other). If we had some prior knowledge of the label spaces, we could use it to define a notion of distance between pairs of labels. However, in the challenging —but common— case where no such knowledge is available, the only information we have about the labels is their occurrence in relation to the feature vectors $x$. Thus, we can take advantage of the fact that we have a meaningful metric in $\mathcal{X}$ and use it to compare labels.

Formally, let $\mathsf{N}_{\mathcal{D}}(y) := \{x \in \mathcal{X} \mid (x, y) \in \mathcal{D}\}$ be the set of feature vectors with label $y$, and let $n_y$ be its cardinality. With this, a distance between two labels $y$ and $y'$ could be defined as the distance between the centroids of $\mathsf{N}_{\mathcal{D}}(y)$ and $\mathsf{N}_{\mathcal{D}}(y')$. But representing the collections $\mathsf{N}_{\mathcal{D}}(y)$ only through their mean is likely too simplistic for real datasets. Ideally, we would represent labels through the *actual distribution* over the feature space that they define, namely, by the map $y \mapsto \alpha_y(X) \triangleq P(X \mid Y = y)$, of which $\mathsf{N}_{\mathcal{D}}(y)$ can be understood as a finite sample. If we use this representation, defining a distance between labels boils down to choosing a divergence between their associated distributions. Here again we argue that OT is an ideal choice, since it: (i) yields a true metric, (ii) is computable from finite samples, which is crucial since the distributions $\alpha_y$ are not available in analytic form, and (iii) is able to deal with sparsely-supported distributions.

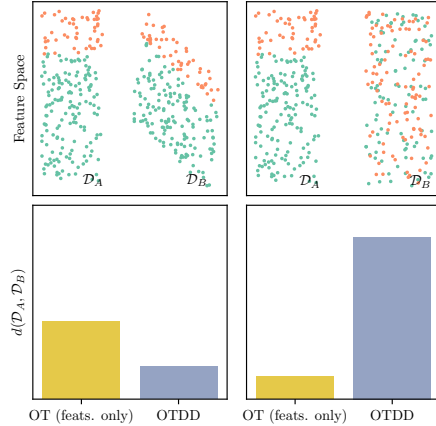

**Figure 2: The importance of labels.** The second pair of datasets are much closer than the first under the usual (label-agnostic) OT distance, while the opposite is true for our (label-aware) distance.

The approach described so far *grounds* the comparison of the $\alpha_y$ distributions to the feature space $\mathcal{X}$, so we can simply use $d_{\mathcal{X}}^p$ as the optimal transport cost, leading to a p-Wasserstein distance between labels: $\mathrm{W}_p^p(\alpha_y, \alpha_{y'})$, and in turn, to the following distance between feature-label pairs:

$$d_{\mathcal{Z}}\big((x, y), (x', y')\big) \triangleq \big(d_{\mathcal{X}}(x, x')^p + \mathrm{W}_p^p(\alpha_y, \alpha_{y'})\big)^{1/p}. \qquad (5)$$

With this notion of distance in $\mathcal{Z}$ at hand, we can finally use optimal transport again to lift this point-wise metric into a distance between measures (and therefore, between datasets):

$$d_{\mathrm{OT}}(\mathcal{D}_A, \mathcal{D}_B) = \min_{\pi \in \Pi(\alpha, \beta)} \int_{\mathcal{Z} \times \mathcal{Z}} d_{\mathcal{Z}}(z, z')^p \pi(z, z'). \tag{6}$$

As proved in Appendix A, this defines a true metric on $\mathcal{P}(\mathcal{Z})$ – the Optimal Transport Dataset Distance (OTDD). Figure 1 illustrates the main aspects of this distance on a simple 2D dataset.

It remains to describe how the distributions $\alpha_y$ are to be represented. One could treat the samples in $\mathsf{N}_{\mathcal{D}}(y)$ as support points of a uniform empirical measure, *i.e.,* $\alpha_y = \sum_{\mathbf{x}^{(i)} \in \mathsf{N}_{\mathcal{D}}(y)} \frac{1}{n_y} \delta_{\mathbf{x}^{(i)}}$, as described in Section 3. In this case, *every* evaluation of (5) would involve solving an OT problem, for a total worst-case $O(n^5 \log n)$ complexity, as shown in §C.1. This might be prohibitive in some settings. For those cases, we propose an alternative approach that relies on representing the $\alpha_y$ as Gaussian distributions, which leads to a simple yet tractable realization of the distance (6). Formally, we model each $\alpha_y$ as a Gaussian $\mathcal{N}(\hat{\mu}_y, \hat{\Sigma}_y)$ whose parameters are the sample mean and covariance of $\mathsf{N}_{\mathcal{D}}(y)$. The main advantage of this approach is that the 2-Wasserstein distance between Gaussians $\mathcal{N}(\mu_\alpha, \Sigma_\alpha)$ and $\mathcal{N}(\mu_\beta, \Sigma_\beta)$ has an analytic form, often known as the Bures-Wasserstein distance:

$$\mathrm{W}_2^2(\alpha, \beta) = \|\mu_\alpha - \mu_\beta\|_2^2 + \mathrm{tr}(\Sigma_\alpha + \Sigma_\beta - 2(\Sigma_\alpha^{\frac{1}{2}} \Sigma_\beta \Sigma_\alpha^{\frac{1}{2}})^{\frac{1}{2}}) \tag{7}$$

where $\Sigma^{\frac{1}{2}}$ is the matrix square root. Furthermore, whenever $\Sigma_\alpha$ and $\Sigma_\beta$ commute, this further simplifies to

$$\mathrm{W}_2^2(\alpha, \beta) = \|\mu_\alpha - \mu_\beta\|_2^2 + \|\Sigma_\alpha^{\frac{1}{2}} - \Sigma_\beta^{\frac{1}{2}}\|_F^2. \tag{8}$$

When using the Bures-Wasserstein distance in $d_{\mathcal{Z}}$ (5), we denote the resulting dataset distance (6) by $d_{\mathrm{OT}\text{-}\mathcal{N}}$, or Bures-OTDD.

Representing label-induced distributions as Gaussians might seem like a heuristic —and potentially, overly coarse— approximation. In cases where the data is first embedded with some complex non-linear mapping (*e. g.,* a neural network, as in our text classification experiments §6.4), there is empirical evidence that the first two moments capture enough relevant information for classification [23]. On the other hand, the following result, a consequence of a bound by Gelbrich [26], shows that we can provably 'sandwich' the exact $d_{\mathrm{OT}}$ by this Gaussian approximation and a trivial and easily computable upper bound (defined in Appendix B):

**Proposition 4.1.** *For any two datasets $\mathcal{D}_A, \mathcal{D}_B$, we have:*

$$d_{\mathrm{OT}\text{-}\mathcal{N}}(\mathcal{D}_A, \mathcal{D}_B) \leq d_{\mathrm{OT}}(\mathcal{D}_A, \mathcal{D}_B) \leq d_{\mathrm{UB}}(\mathcal{D}_A, \mathcal{D}_B) \tag{9}$$

*where $d_{\mathrm{UB}}$ is a distribution-agnostic OT upper bound. Furthermore, the first two distances are equal if all the label distributions $\alpha_y$ are Gaussian or elliptical (i. e., $d_{\mathrm{OT}\text{-}\mathcal{N}}$ is exact in that case).*

In the next section, we compare $d_{\mathrm{OT}}$ and $d_{\mathrm{OT}\text{-}\mathcal{N}}$ in terms of their computational complexity, and in Section 6.1 we perform ablations comparing these and other baselines, including the lower and upper bounds of Proposition 4.1, in controlled experimental settings.

## 5    Computational Considerations

Since our goal in this work is to leverage dataset distances for tasks like transfer learning in realistic (*i. e.,* large) machine learning datasets, scalability is crucial. Indeed, most compelling use cases of *any* notion of distance between datasets will involve computing it repeatedly on very large samples. While estimation of Wasserstein —and more generally, optimal transport— distances is known to be computationally expensive in general, in Section 3 we mentioned how entropy regularization can be used to trade-off accuracy for runtime. Recall that both the general and Gaussian versions of the dataset distance proposed in Section 4 involve solving optimal transport problems (though the latter, owing to the closed-form solution of subproblem (7), only requires optimization for the global problem). Therefore, both of these distances benefit from approximate OT solvers.

But further speed-ups are possible. For $d_{\mathrm{OT}\text{-}\mathcal{N}}$, a simple and fast implementation can be obtained if (i) the metric in $\mathcal{X}$ coincides with the ground metric in the transport problem on $\mathcal{Y}$, and (ii) all covariance matrices commute. While (ii) will rarely occur in practice, one could use a diagonal approximation to the covariance, or with milder assumptions, simultaneous matrix diagonalization [18]. In either case, using the simplification in (8), the pointwise distance $d(z, z')$ can be computed

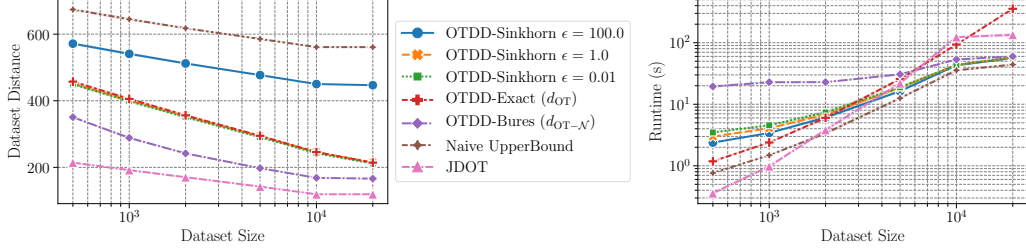

**Figure 3:** Comparison of variants of OTDD using different methods to compute the label distance $d_{\mathcal{Y}}$, and other baselines. The distances are computed between subsets of MNIST drawn independently.

by creating augmented representations of each dataset, whereby each pair $(x, y)$ is represented as a stacked vector $\tilde{x} := [x; \mu_y; \text{vec}(\Sigma_y^{1/2})]$ for the corresponding label mean and covariance. Then, $\|\tilde{x} - \tilde{x}'\|_2^2 = d_{\mathcal{Z}}(x, y; x', y')^2$ for $d_{\mathcal{Z}}$ as defined in Eq. (5). Therefore, in this case the OTDD can be immediately computed using an off-the-shelf OT solver on these augmented datasets. While this approach is appealing computationally, here instead we focus on a exact version that does not require diagonal or commuting covariance approximations, and leave empirical evaluation of this approximate approach for future work.

The steps we propose next are motivated by the observation that, unlike traditional OT distances for which the cost of computing pair-wise distance is negligible compared to the complexity of the optimization routine, in our case the latter dominates, since it involves computing multiple OT distances itself. In order to speed up computation, we first pre-compute and store in memory all label-to-label pairwise distances $d(\alpha_y, \alpha_{y'})$, and retrieve them on-demand during the optimization of the outer OT problem. For $d_{\text{OT-}\mathcal{N}}$, computing the label-to-label distances $d(\mathcal{N}(\hat{\mu}_y, \hat{\Sigma}_y), \mathcal{N}(\hat{\mu}_{y'}, \hat{\Sigma}_{y'}))$ is dominated by the cost of computing matrix square roots, which if done exactly involves a full eigendecomposition. Instead, it can be computed approximately using the Newton-Schulz iterative method [29, 44]. Besides runtime, loading all examples of a given class to memory (to compute means and covariances) might be infeasible for large datasets (especially if running on GPU), so we instead use a two-pass stable online batch algorithm to compute these statistics [10].

The following result, proven in the Supplement §C, summarizes the time complexity of our two distances and sheds light on the trade-off between precision and efficiency they provide.

**Theorem 5.1.** *For datasets of size $n$ and $m$, with $p$ and $q$ classes, dimension $d$, and maximum class size $\mathfrak{n}$, both $d_{\text{OT}}$ and $d_{\text{OT-}\mathcal{N}}$ incur in a cost of $O(nm \log(\max\{n, m\})\tau^{-3})$ for solving the outer OT problem $\tau$-approximately, while the worst-case complexity for computing the label-to-label pairwise distances (5) is $O(nm(d + \mathfrak{n}^3 \log \mathfrak{n} + d\mathfrak{n}^2))$ for $d_{\text{OT}}$ and $O(nmd + pqd^3 + d^2\mathfrak{n}(p + q))$ for $d_{\text{OT-}\mathcal{N}}$.*

For small to medium-sized datasets, computing the exact $d_{\text{OT}}$ is feasible and might even be faster than computing $d_{\text{OT}-\mathcal{N}}$, *e. g.,* when $d \gg n$, in which case $d^3$ dominates the complexity in Theorem 5.1. This can be observed in practice too (Fig. 3). For very large datasets ($n \gg d$), the cost of computing pairwise distances will often dominate. For example, if $n = m$ and the largest class size is $O(n)$, this step becomes $O(n^5 \log n)$ —prohibitive for large datasets— for $d_{\text{OT}}$ but only $O(n^2 d + d^3)$ for $d_{\text{OT-}\mathcal{N}}$. In such cases, $d_{\text{OT-}\mathcal{N}}$ might be the only viable option.

## 6 Experiments

### 6.1 Asymptotics and Runtime for Variations of the OTDD

In our first set of experiments, we investigate the behavior and quality of the dataset distance proposed here for variations on how the label-to-label distance $d_{\mathcal{Y}}$ is computed. Recall that the two main approaches do so proposed here are to compute it as an exact distance between empirical measures (leading to $d_{\text{OT}}$) or using a Gaussian approximation ($d_{\text{OT-}\mathcal{N}}$). For the former, we discussed in Section 5 possible speed-ups by solving this inner OT problem approximately using a Sinkhorn divergence (4). We compare these three variants (exact, bures, sinkhorn), along with three other baselines: the upper bound of Theorem 5.1, the means-only approximation of the label distributions discussed in Section 4, and the JDOT approach [14] that uses a classification loss. Note that this last approach is

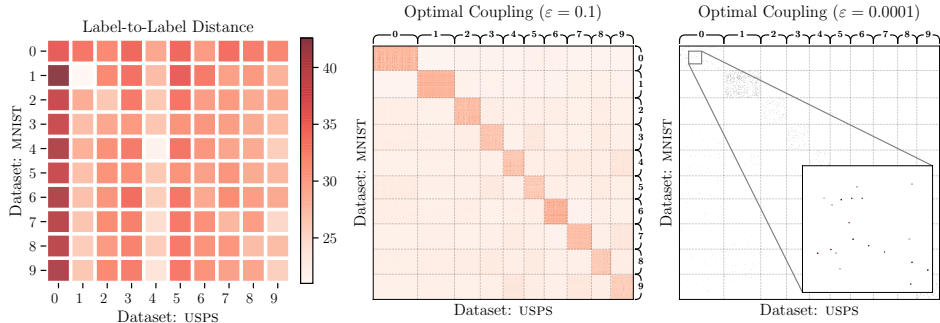

**Figure 4: Dataset Distance between MNIST and USPS.** **Left**: The label Wasserstein distances — computed without knowledge of the relation between labels across domains— recover expected relations between these classes. **Center/Right**: The optimal coupling $\pi^*$ for different regularization levels exhibits a block-diagonal structure, indicating class-coherent matches across domains.

not directly comparable, as its scaling depends on the classification loss used, and is only applicable to settings where the two datasets have identical label sets. For our experiments, we take independent samples of MNIST of increasing size, and compute distance with all these methods. The results (Figure 3) exhibit various interesting phenomena. First, consistent with Proposition 4.1, the exact $d_{\text{OT}}$ is bounded by $d_{\text{UB}}$ and the approximate $d_{\text{OT-}\mathcal{N}}$, with the latter being remarkably tight, particularly as dataset size grows. The Sinkhorn-based versions of OTDD interpolate between these two bounds. Finally, as predicted by Theorem 5.1, in this case the exact OTDD is in fact faster to compute that the approximate $d_{\text{OT-}\mathcal{N}}$ for small dataset sizes ($\lesssim$5K samples), and again, using Sinkhorn for the inner OT problem allows us to interpolate between these two regimes.

## 6.2 Dataset Selection for Transfer Learning

In this section, we test whether the OTDD can provide a learning-free criterion on which to select a source dataset for transfer learning. We start with a simple domain adaptation setting, using USPS, MNIST [36] and three of its extensions: Fashion-MNIST [54], KMNIST [11] and the `letters` split of EMNIST [12]. All datasets consist of 10 classes, except EMNIST, for which the selected split has 26 classes. Throughout this section, we use a simple LeNet-5 neural network (two convolutional layers, three fully connected ones) with ReLU activations. When carrying out adaptation, we freeze the convolutional layers and fine-tune only the top three layers.

We first compute all pairwise OTDD distances (Fig 5). For the example of $d_{\text{OT-}\mathcal{N}}(\text{MNIST}, \text{USPS})$, Figure 4 illustrates two key components of the distance: the label-to-label distances $d_{\mathcal{Y}}$ (left) and the optimal coupling $\pi^*$ obtained for two levels of entropy regularization $\varepsilon$ (center, right). The diagonal elements of the first plot (*i.e.,* distances between congruent digit classes) are overall relatively smaller than off-diagonal elements. Interestingly, the 0 class of USPS appears remarkably far from *all* MNIST digits under this metric. On the other hand, most correspondences lie along the (block) diagonal of $\pi^*$, which shows the dataset distance is able to infer class-coherent correspondences across them. Despite both consisting of

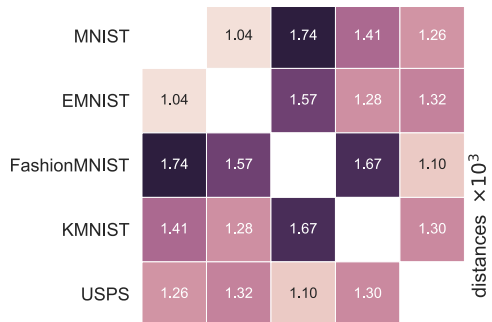

**Figure 5:** Pairwise OT Distances

digits, MNIST and USPS are not the closest among these datasets according to the OTDD, as Figure 5 shows. The closest pair is instead (MNIST, EMNIST), while Fashion-MNIST appears far from all others, particularly MNIST.

We test the robustness of the distance by computing it repeatedly for varying sample sizes. The results (Fig. 9, Appendix G) show that the distance converges towards a fixed value as sample sizes grow, but interestingly, small sample sizes for USPS lead to wider variability, suggesting that this dataset itself is more heterogeneous than MNIST.

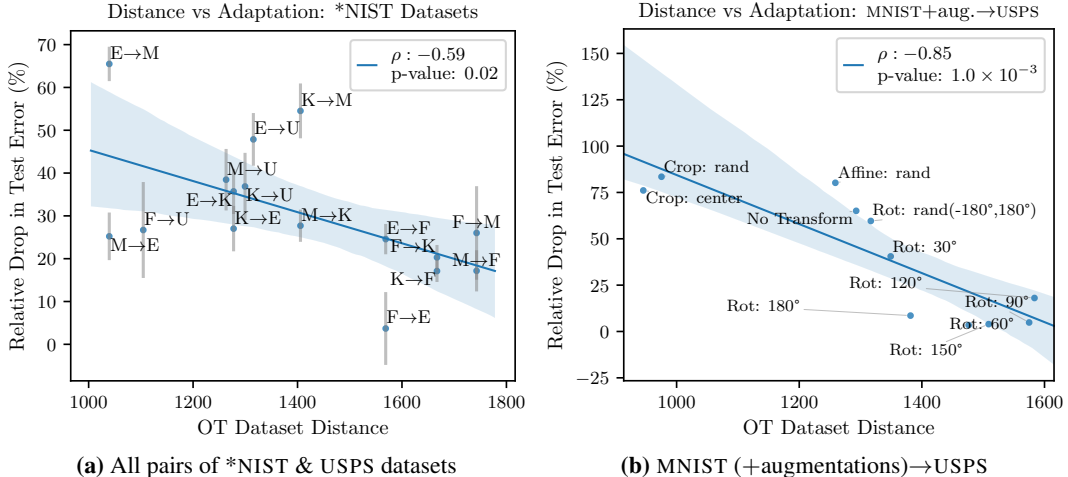

**(a)** All pairs of *NIST & USPS datasets  **(b)** MNIST (+augmentations)→USPS

**Figure 6:** Comparison of the OT dataset distance and transferability between *NIST datasets.

Next, we compare the OTDD against the *transferability* between datasets, *i. e.,* the gain in performance from using a model pretrained on the source domain and fine-tuning it on the target domain, compared to not pretraining. To make these numbers comparable across dataset pairs, we report *relative* drop in classification error brought by adaptation: $\mathcal{T}(\mathcal{D}_S \to \mathcal{D}_T) = 100 \times \frac{\text{error}(\mathcal{D}_S \to \mathcal{D}_T) - \text{error}(\mathcal{D}_T)}{\text{error}(\mathcal{D}_T)}$.

We run the adaptation task 10 times with different random seeds for each pair of datasets, and compare $\mathcal{T}$ against their distance. The strong and significant correlation between these (Fig. 6a) shows that the OTDD is highly predictive of transferability across these datasets. In particular, EMNIST led to the best adaptation to MNIST, justifying the —initially counter-intuitive— value of the OTDD. Repeating this experiment with ablated versions of the OTDD shows that using both feature and label information, and modeling second-order moments, are crucial to achieve this strength of correlation with transferability (Figure 8, Appendix F).

### 6.3 Distance-Driven Data Augmentation

Data augmentation —*i. e.,* applying carefully chosen transformations on a dataset to enhance its quality and diversity— is another key aspect of transfer learning that has substantial empirical effect on the quality of the transferred model yet lacks principled guidelines. Here, we investigate if the OTDD could be used to compare and select among possible augmentations.

For a fixed source-target dataset pair, we generate copies of the source data with various transformations applied to it, compute their distance to the target dataset, and compare to transfer accuracy as before. We present results for a small-scale (MNIST→USPS) and a larger-scale (Tiny-ImageNet→CIFAR-10) setting. The transformations we use on MNIST consist of rotations by a fixed degree $[30°, \ldots, 180°]$, random rotations $(-180°, 180°)$, random affine transformations, center- and random-crops. For Tiny-ImageNet we randomly vary brightness, contrast, hue and saturation. The models used are respectively the LeNet-5 and a ResNet-50 (training details in Appendix §E). The results in both of these settings (Figures 6b and 7a) show, again, a strong significant correlation between these two. A reader familiar with the MNIST and USPS datasets will not be surprised by the fact that cropping images from the former leads to substantially better performance on the latter, while most rotations degrade transferability.

### 6.4 Transfer Learning for Text Classification

Natural Language Processing (NLP) has recently seen a profound impact from large-scale transfer learning, largely driven by the availability of off-the-shelf large language models pre-trained on massive amounts of data [21, 45, 47]. While natural language inherently lacks the fixed-size continuous vector representation required by our framework to compute pointwise distances, we can take advantage of these pretrained models to embed sentences in a vector space with rich geometry. In our experiments, we first embed the sentences of every dataset using the (base) BERT model [21], and then compute the OTDD on these embedded datasets.

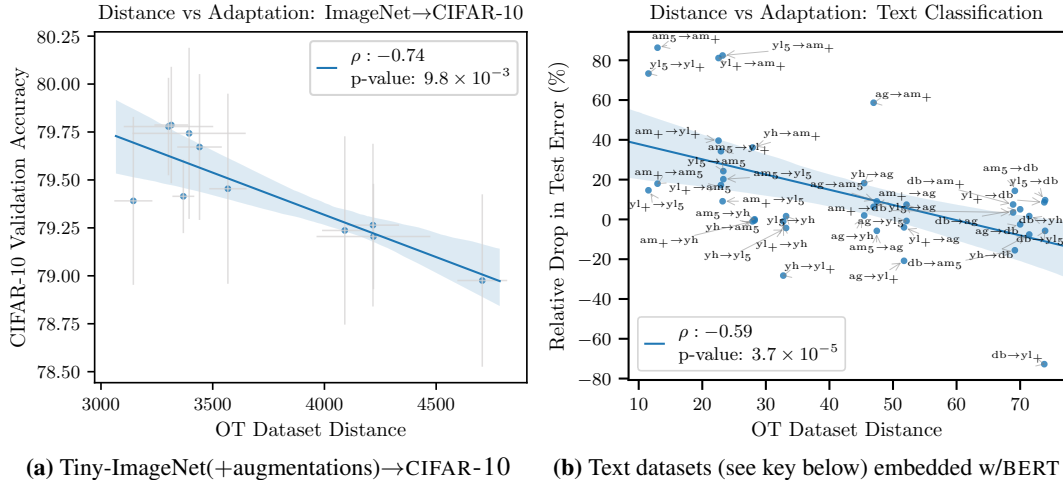

**(a)** Tiny-ImageNet(+augmentations)→CIFAR-10

**(b)** Text datasets (see key below) embedded w/BERT

**Figure 7:** Comparison of the OT dataset distance and transferability in two large-scale settings.

We focus on the problem of sentence classification, and consider the following datasets[2] by Zhang et al. [56]: AG news (ag), DBPedia (db), Yelp Reviews with 5-way classification ($yl_5$) and binary polarity ($yl_+$) label encodings, Amazon Reviews with 5-way classification ($am_5$) and binary polarity ($am_+$) label encodings, and YAHOO Answers (yh). We provide details for all these datasets in Appendix D.

As before, we simulate a challenging adaptation setting by keeping only 100 examples per target class. For every pair of datasets, we first fine-tune the BERT model using the entirety of the source domain data, after which we fine-tune and evaluate on the target domain. Figure 7b shows that the OT dataset distance is highly correlated with transferability in this setting too. Interestingly, adaptation often leads to drastic degradation of performance in this case, which suggests that off-the-shelf BERT is on its own powerful and flexible enough to initialize many of these tasks, and therefore choosing the wrong domain for initial training might destroy some of that information.

## 7    Discussion

We have shown that the notion of distance between datasets proposed in this work is scalable and flexible, all the while offering appealing theoretical properties and interpretable comparisons. To allow for scaling up to very large datasets, we have proposed approximate versions of the OTDD, which despite trading off quality for runtime, often perform almost as well as the exact one in practice. Naturally, any refinement over these approximations would only further mitigate this trade-off. In terms of applications, our results on transfer learning scenarios demonstrate the practical utility of this novel distance, and are consistent with prior work showing that adaptation is most likely —and often *only*— successful if the domains are not too different [41].

There are many natural extensions and varitions of this distance. Here we assumed that the datasets were defined on feature spaces of the same dimension, but one could instead leverage a relational notion such as the Gromov-Wasserstein distance [43] to compute the distance between datasets whose features and not directly comparable. On the other hand, our efficient implementation relies on modeling groups of points with the same label as Gaussian distributions. This could naturally be extended to more general distributions for which the Wasserstein distance either has an analytic solution or at least can be computed efficiently, such as elliptic distributions [44], Gaussian mixture models [19], certain Gaussian Processes [40], or tree metrics [35].

In this work, we purposely excluded two key aspects of any learning task from our notion of distance: the loss function and the predictor function class. While we posit that it is crucial to have a notion of distance that is independent of these choices, it is nevertheless appealing to ask whether our distance could be extended to take those into account, ideally involving minimal training. Exploring different avenues to inject such information into this framework will be the focus of our future work.

## Broader Impact

A notion of distance is such a basic and fundamental concept that it is most often used as a primitive from which other tools and methods derive utility. In the specific case of the dataset distance we propose here, it would most likely be used as tool within a machine learning pipeline. Thus, by its very nature, the prospect of potential impact of this work is broad enough to essentially encompass most settings where machine learning is used. In this statement, we focus on aspects that are immediate, tractable, and precise enough to be discussed constructively in this format.

Perhaps the most immediate impact of this work could be through its application in transfer learning. Improvements in this paradigm can have a myriad outcomes, ranging from societal to environmental, both within and beyond the machine learning community. Among potential beneficial outcomes, one that stands out is the environmental impact of making transfer learning more efficient by providing guidance as to what resources to use for pretraining (§6.2) or choosing optimal data augmentations (§6.3). This would be particularly relevant for NLP, where the carbon footprint of models has grown exponentially in recent years, driven largely by pretraining of very large models on massive datasets [49]. Another beneficial outcome of this specific use of the distance proposed in this work rests on the intuition that more efficient transfer learning would could erode or mitigate economic barriers that currently limit large-scale data pretraining and adaptation to resource-rich entities and institutions. However, work studying the impact of improved data and method efficiency has pointed out that this intuition is perhaps too optimistic, as there are various unexpected yet feasible negative collateral consequences of increased efficiency, *e. g.,* in terms of privacy, data markets and misuse [51].

We next highlight a few potential failure modes of this work. The modeling approximations used here to make this notion of distance efficiently computable, in particular the use of Gaussian distribution for modeling same-class collections, might prove too unrealistic in some datasets, leading to unreliable distance estimation. This, of course, could have negative impact on downstream applications that would rely on this distance as a sub-component, especially so given how deeply embedded within an ML pipeline it would be. In order to mitigate such impact, we suggest the practitioner verify how realistic these modeling assumptions are for the application at hand. On the other hand, despite the limited number of hyperparameters the computation of this distance relies on, inadequate choices for these (*e. g.,* the entropy regularization parameter $\varepsilon$) might nevertheless lead to unreliable or imprecise results. Again, care should be taken in test the validity of the parameters, ideally running sanity-checks on identical or near-identical datasets to corroborate that the results are sensible.

## Acknowledgments and Disclosure of Funding

D.A.M. and N.F. were employed by Microsoft corporation while performing this work.

## Footnotes

[1]Despite its name, this discrepancy is not a distance in general.

[2]Available via the `torchtext` library.

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
