[Supplementary Material]

# A    OTDD is a True Distance

**Proposition A.1.** $d_{\text{OT}}(\mathcal{D}_A, \mathcal{D}_B)$ *defines a valid metric on* $\mathcal{P}(\mathcal{X} \times \mathcal{P}(\mathcal{X}))$ *the space of measures over feature and label-distribution pairs.*

*Proof.* Whenever the cost function used is a metric in a given space $\mathcal{X}$, the optimal transport problem itself defines a distance (the Wasserstein distance) on $\mathcal{P}(\mathcal{X})$ [52, Chapter 6]. Therefore, it suffices to show that the cost function $d_{\mathcal{Z}}$ defined in Eq. (5) is indeed a distance. Clearly, it is symmetric because both $d_{\mathcal{X}}$ and $\text{W}_p$ are. In addition, since both of these are distances:

$$d_{\mathcal{Z}}(z, z') = 0 \Leftrightarrow d_{\mathcal{X}}(x, x') = 0 \wedge \text{W}_p(\alpha_y, \alpha'_y) = 0 \Leftrightarrow x = x', \ \alpha_y = \alpha'_y \Leftrightarrow z = z'$$

Finally, we have that

$$\begin{aligned}
d_{\mathcal{Z}}(z_1, z_3) &= \left( d_{\mathcal{X}}(x_1, x_3)^p + \text{W}_p(\alpha_{y_1}, \alpha_{y_3})^p \right)^{\frac{1}{p}} \\
&\leq \left( d_{\mathcal{X}}(x_1, x_2)^p + d_{\mathcal{X}}(x_2, x_3)^p + \text{W}_p(\alpha_{y_1}, \alpha_{y_2})^p + \text{W}_p(\alpha_{y_2}, \alpha_{y_3})^p \right)^{\frac{1}{p}} \\
&= \left( d_{\mathcal{Z}}(z_1, z_2)^p + d_{\mathcal{Z}}(z_2, z_3)^p \right)^{\frac{1}{p}} = d_{\mathcal{Z}}(z_1, z_2) + d_{\mathcal{Z}}(z_2, z_3)
\end{aligned}$$

where the last step is an application of Minkowski's inequality. Hence, $d_{\mathcal{Z}}$ satisfies the triangle inequality, and therefore it is a metric on $\mathcal{Z} = \mathcal{X} \times \mathcal{P}(\mathcal{X})$. We therefore conclude that the value of the optimal transport (6) that uses this metric as a cost function is a distance itself. $\square$

# B    Proof of Proposition 4.1

Proposition 4.1 is a direct extension of the following well-known bound for the 2-Wasserstein distance due to Gelbrich [26]:

**Lemma B.1** (Gelbrich bound). *Suppose* $\alpha, \beta \in \mathcal{P}(\mathbb{R}^d)$ *are any two measures with mean vectors* $\mu_\alpha, \mu_\beta \in \mathbb{R}^d$ *and covariance matrices* $\Sigma_\alpha, \Sigma_\beta \in \mathbb{S}_+^d$ *respectively. Then,*

$$\text{W}_2^2\big(\mathcal{N}(\mu_\alpha, \Sigma_\alpha), \mathcal{N}(\mu_\beta, \Sigma_\beta)\big) \leq \text{W}_2^2(\alpha, \beta) \tag{10}$$

*where* $\text{W}_2^2\big(\mathcal{N}(\mu_\alpha, \Sigma_\alpha), \mathcal{N}(\mu_\beta, \Sigma_\beta)\big)$ *is as in Eq. (7).*

Obtaining an upper bound is trivial, noting that for any two measures $\alpha, \beta$,

$$\text{W}_2^2(\alpha, \beta) = \|\mu_\alpha - \mu_\beta\|_2^2 + \text{tr}(\Sigma_\alpha + \Sigma_\beta) - 2 \max_{\pi \in \Pi} \text{tr}(\Sigma_\pi) \leq \|\mu_\alpha - \mu_\beta\|_2^2 + \text{tr}(\Sigma_\alpha + \Sigma_\beta). \tag{11}$$

Let $d_{\text{UB}}(\mathcal{D}_A, \mathcal{D}_B)$ denote the OT distance obtained by using the cost $d_{\mathcal{Z}}^2(z, z') = d_{\mathcal{X}}(x, x')^2 + \|\mu_y - \mu_{y'}\|_2^2 + \text{tr}(\Sigma_y + \Sigma_{y'})$. Then, for our setting, we have:

**Proposition 4.1.** *For any two datasets* $\mathcal{D}_A, \mathcal{D}_B$, *we have:*

$$d_{\text{OT-}\mathcal{N}}(\mathcal{D}_A, \mathcal{D}_B) \leq d_{\text{OT}}(\mathcal{D}_A, \mathcal{D}_B) \leq d_{\text{UB}}(\mathcal{D}_A, \mathcal{D}_B) \tag{12}$$

*where* $d_{\text{UB}}$ *is a distribution-agnostic OT upper bound. Furthermore, the first two distances are equal if all the label distributions* $\alpha_y$ *are Gaussian or elliptical (i. e.,* $d_{\text{OT-}\mathcal{N}}$ *is exact in that case).*

*Proof.* In the notation of Section 3, Lemma B.1 implies that for every feature-label pairs $z = (x, y)$ and $z' = (x', y')$, we have:

$$d_{\mathcal{X}}(x, x')^2 + \text{W}_2^2\big(\mathcal{N}(\mu_y, \Sigma_y), \mathcal{N}(\mu_{y'}, \Sigma_{y'})\big) \leq d_{\mathcal{X}}(x, x')^2 + \text{W}_2^2(\alpha_y, \alpha_{y'}), \tag{13}$$

and therefore

$$\int d_{\mathcal{Z}}(z, z')^2 \, d\pi \leq \int d_{\mathcal{Z}}(z, z')^2 \, d\pi \tag{14}$$

for every coupling $\pi \in \Pi(\alpha, \beta)$. In particular, for the minimizing $\pi^*$, we obtain that

$$d_{\text{OT-}\mathcal{N}}(\mathcal{D}_A, \mathcal{D}_B) \leq d_{\text{OT}}(\mathcal{D}_A, \mathcal{D}_B) \tag{15}$$

We obtain the upper bound analogously.

Clearly, Gelbrich's bound holds with equality when $\alpha$ and $\beta$ are indeed Gaussian. More generally, equality is attained for elliptical distributions with the same density generator [34]. This immediately implies equality of the first two terms in equation (15) in that case. $\square$

## C   Time Complexity Analysis

For the analyses in this section, assume that $\mathcal{D}_S$ and $\mathcal{D}_T$ respectively have $n$ and $m$ labeled examples in $\mathbb{R}^d$ and $k_s, k_t$ classes. In addition, let $\mathsf{N}_{\mathcal{D}}^S(i) := \{x \in \mathcal{X} \mid (x, y = i) \in \mathcal{D}\}$ be the subset of examples in $\mathcal{D}_S$ with label $i$, and define analogously $\mathsf{N}_{\mathcal{D}}^T(j)$. The denote the cardinalities of these subsets as $n_s^i \triangleq |\mathsf{N}_s^{(i)}|$ and analogously for $n_t^j$.

Direct computation of the distance (5) involves two main steps:

(i) computing pairwise pointwise distances (each requiring solution of a label-to-label OT sub-problem), and

(ii) a global OT problem between the two samples.

Step (ii) is identical for both the general distance $d_{\mathrm{OT}}$ and its Gaussian approximation counterpart $d_{\mathrm{OT}\text{-}\mathcal{N}}$, so we analyze it first. This is an OT problem between two discrete distributions of size $n$ and $m$, which can be solved exactly in $O\big((n+m)nm\log(nm)\big)$ using interior point methods or Orlin's algorithm for the uncapacitated min cost flow problem [46]. Alternatively, it can be solved $\tau$-approximately in $O(nm\log(\max\{n, m\})\tau^{-3})$ time using the Sinkhorn algorithm [3].

We next analyze step (i) individually for the two OTDD versions. Combined, they provide a proof of Theorem 5.1.

### C.1   Pointwise distance computation for $d_{\mathrm{OT}}$

Consider a single pair of points, $(x, y = i) \in \mathcal{D}_A$ and $(x', y' = j) \in \mathcal{D}_B$. Evaluating $\|x - x'\|$ has $O(d)$ complexity, while $W(\alpha_y, \beta_{y'})$ is an $n_s^i \times n_t^j$ OT problem which itself requires computing a distance matrix (at cost $O(n_s^i n_t^j d)$), and then solving the OT problem, which as discussed before, be done exactly in $O\big((n_s^i + n_t^j)n_s^i n_t^j \log(n_s^i + n_t^j)\big)$ or $\tau$-approximately in $O(n_s^i n_t^j \log(\max\{n_s^i, n_t^j\})\tau^{-3})$.

For simplicity, let us denote $\mathfrak{n}_s = \max_i n_s^i$, and $\mathfrak{n}_t = \max_j n_t^j$ the size of the largest label cluster in each dataset, and $\mathfrak{n} = \max\{\mathfrak{n}_s, \mathfrak{n}_t\}$ the overall largest one. Using these, and combining all of the above, the overall worst case complexity for the computation of the $n \times m$ pairwise distances can be expressed as

$$O\big(nm(d + \mathfrak{n}^3 \log \mathfrak{n} + d\mathfrak{n}^2)\big), \tag{16}$$

which is what we wanted to show.

$\square$

### C.2   Pointwise distance computation for $d_{\mathrm{OT}\text{-}\mathcal{N}}$

As before, consider a pair of points $(x, y = i) \in \mathcal{D}_A$ and $(x', y' = j) \in \mathcal{D}_B$ whose cluster sizes are $n_s^i$ and $n_t^j$ respectively. As mentioned in Section 5, for $d_{\mathrm{OT}\text{-}\mathcal{N}}$ we first compute all the per-class means and covariance matrices. This step is clearly dominated by latter, which is $O(d^2 n_s^i)$.[3] Considering all labels from both datasets, this amounts to a worst-case complexity of $O\big(d^2(k_s\mathfrak{n}_s + k_t\mathfrak{n}_t)\big)$.

Once the means and covariances have been computed, we precompute all the $k_s \times k_t$ pair-wise label-to-label distances $W_2(\alpha_y, \beta_{y'})$ using Eq. (7). This computation is dominated by the matrix square roots. If done exactly, these involve a full eigendecomposition, at cost $O(d^3)$, so the total cost for this step is $O(k_s k_t d^3)$.

Finally, while computing the pairwise distance, we will incur in $O(nmd)$ to obtain $\|x - x'\|$. Putting all of these together, and replacing $\mathfrak{n}_s, \mathfrak{n}_t$ by $\mathfrak{n}$, we obtain a total cost for precomputing all the point-wise distances of:

$$O(nmd + k_s k_t d^3 + d^2\mathfrak{n}(k_s + k_t)),$$

which concludes the proof.

$\square$

# D Dataset Details

Information about all the datasets used, including references, are provided in Table 1.

| Dataset | Input Dimension | Number of Classes | Train Examples | Test Examples | Source |
|---|---|---|---|---|---|
| USPS | $16 \times 16^{*}$ | 10 | 7291 | 2007 | [30] |
| MNIST | $28 \times 28$ | 10 | 60K | 10K | [36] |
| EMNIST (letters) | $28 \times 28$ | 26 | 145K | 10K | [12] |
| KMNIST | $28 \times 28$ | 10 | 60K | 10K | [11] |
| FASHION-MNIST | $28 \times 28$ | 10 | 60K | 10K | [54] |
| TINY-IMAGENET | $64 \times 64^{\ddagger}$ | 200 | 100K | 10K | [20] |
| CIFAR-10 | $32 \times 32$ | 10 | 50K | 10K | [33] |
| AG news | $768^{\dagger}$ | 4 | 120K | 7.6K | [56] |
| DBPedia | $768^{\dagger}$ | 14 | 560K | 70K | [56] |
| YELPREVIEW (Polarity) | $768^{\dagger}$ | 2 | 560K | 38K | [56] |
| YELPREVIEW (Full Scale) | $768^{\dagger}$ | 5 | 650K | 50K | [56] |
| AMAZONREVIEW (Polarity) | $768^{\dagger}$ | 2 | 3.6M | 400K | [56] |
| AMAZONREVIEW (Full Scale) | $768^{\dagger}$ | 5 | 3M | 650K | [56] |
| YAHOO ANSWERS | $768^{\dagger}$ | 10 | 1.4M | 60K | [56] |

**Table 1:** Summary of datasets used. $*$: we rescale the USPS digits to $28 \times 28$ for comparison to the *NIST datasets. $\ddagger$: we rescale Tiny-ImageNet to $32 \times 32$ for comparison to CIFAR-10. $\dagger$: for text datasets, variable-length sentences are embedded to fixed-dimensional vectors using BERT.

# E Optimization and Training Details

For the adaptation experiments on the *NIST datasets, we use a LeNet-5 architecture with ReLU non-linearities trained for 20 epochs using ADAM with learning rate $1 \times 10^{-3}$, weight decay $1 \times 10^{-6}$, and fine-tuned for 10 epochs on the target domain(s) using the same optimization parameters.

For the Tiny-ImageNet to CIFAR-10 adaptation results, we use a ResNet-50 trained for 300 epochs using SGD with learning rate 0.1 momentum 0.9 and weight decay $1 \times 10^{-4}$ It was fine-tuned for 30 epochs on the target domain using SGD with same parameters except 0.01 learning rate. We discard pairs for which the variance on adaptation accuracy is beyond a certain threshold.

For the text classification experiments, we use a pretrained BERT architecture (the `bert-base-uncased` model of the `transformers`[4] library). We first embed all sentences using this model. Then, for each pair of source/target domains, we first fine-tune using ADAM with learning rate $2 \times 10^{-5}$ for 10 epochs on the full source domain data, and the fine-tune on the restricted target domain data with the same optimization parameters for 2 epochs.

Our implementation of the OTDD relies on the `pot`[5] and `geomloss`[6] python packages.

# F Ablation Experiments on Dataset Selection for Transfer Learning

We repeat the experimental setting of Section 6.2, now using three ablated versions of the OTDD: one which completely ignores the labels (*i.e.,* uses $d_{\mathcal{Z}} = d_{\mathcal{X}}$), one that completely ignores the features ($d_{\mathcal{Z}} = d_{\mathcal{Y}}$), and one that uses a means-only comparison of the label-induced distributions, that is, takes $d_{\mathcal{Y}}(y, y') = \|\mu_y - \mu_y\|$, which can be seen as using a first-order moment approximation of the Bures-Wasserstein distance. Comparing Figure 8 to Figure 6a, we see that both feature and label information is crucial for the OTDD to be predictive of transferability, although, interestingly, dropping the features is not as detrimental, probably because there is already substantial information about these encoded implicitly in the label distributions. On the other hand, the poor performance of the means-only distance shows that second order moment information is crucial.

**Figure 8:** Comparison of ablated versions of OTDD for transferability prediction.

# G  Robustness of the Distance

**Figure 9: Robustness Analysis**: distances computed on subsets of varying size (rows: MNIST, columns: USPS), over 10 random repetitions, for two values of the regularization parameter $\varepsilon$.

## Footnotes

[3]technically, this would be $O(d^\omega n_s^i)$ where $\omega$ is the coefficient of matrix multiplication, but we take $\omega = 3$ for simplicity.

[4] huggingface.co/transformers/

[5] pot.readthedocs.io/en/stable/

[6] www.kernel-operations.io/geomloss/