[Reviews · NeurIPS 2020]

Review 1

Summary and Contributions: The notion of task similarity is of crucial importance in machine learning. It is important in case of domain adaptation, transfer learning, and in those settings where it may be beneficial to pass information across tasks. The goal of the paper is to define a notion of distance between datasets that is model agnostic, does not involve training and has theoretical justification. The distance proposed relies on hierachical optimal transport and does not need any notion of distance between pairs of *labels*. Experiments are shown to empirical evaluate the distance and prove that it correlates well with transfer learning across various datasets and settings. The contributions of the work can be summarized as follows: - the work introduces two similar notions of distances between datasets d_{OT-\cal{N}} and d_{OT}, based on different model assumptions; - it provides algorithms to scale up computation of this distance for large datasets; - it provides empirical evidence that the distance is informative on whether transfer learning across domains is successful.

Strengths: The paper is overall clear and easy to read. The topic is definitely interesting and relevant to machine learning community. The definitions of d_{OT} and its variant d_{OT-N} are independent of any notion of similarity between labels, which makes the proposed notion of distance quite general. The experiments range across a few applications, providing a good empirical evaluation of the method.

Weaknesses: - Regarding the definition of the distance: while I appreciate that this definition allows to deal with general settings, it is not very clear to me how informative the conditional distribution on the labels is. Besides providing some empirical evidence, some in-depth analysis of why it is theoretically principled even in specific settings would be good. - Since transfer learning seems to be a very important and direct application of the distance, having a theoretical evidence / treatment of how the distance is related to the performance would improve the conveyed message.

Correctness: To the best of my knowledge I find the claims and methodology correct.

Clarity: Overall the paper is clear and well written.

Relation to Prior Work: The relation with previous work is well addressed and it is clearly discussed how the work differs from previous literature.

Reproducibility: Yes

Additional Feedback: I would like the author to comment on what is presented in the weaknesses section if possible. Also, - Why entropic OT is not used in the definition in the first place? As pointed out in section 5 on 'Computational Consideration', scalability is crucial for applications of the proposed distance and the authors propose to rely on entropy regularization. Minor comments and typos: - there are \epsilon and \varepsilon in eq(3) ############### AFTER REBUTTAL ############### After reading other reviewers' opinions and the authors' feedback I keep my score unchanged. In line 28 of the feedback the authors say that 'Gaussian approx is faster than Sinkhorn' but this is in contradiction with Fig 1 (right). While I believe that the work is interesting and relevant, I agree with other reviewers on the need of baselines and I hope this is taken into account in the final version.


Review 2

Summary and Contributions: This paper proposes a new distance between classification datasets based on Optimal transport. The key idea is the definition of the OT ground cost between feature-label pairs (x, y) using a combination of a metric on the features space and the Bures-Wasserstein between conditional distributions of the labels (approximated with Gaussians).

Strengths: - The core idea is intuitive and simple to explain - Various experiments were performed to illustrate and showcase the benefits of OTDD

Weaknesses: - No benchmarks / baseline comparisons / sanity checks

Correctness: The claims made in this paper are justified, the conducted experiments are thorough and nicely illustrate the main message with a detailed dicussion on computational aspects.

Clarity: Good writing and clear presentation.

Relation to Prior Work: Yes

Reproducibility: Yes

Additional Feedback: ###### POST REBUTTAL After reading the author's response, I increased my score by 1. I believe the general idea of using conditional distributions to compare datasets with no prior training / modeling assumptions is interesting and could lead to potentially interesting future research. Here is why I still think this is not a clear accept, and I hope these remarks will be addressed in the final version: 1) The experiments that were conducted in the paper were very clear and well illustrated, I expect that the naive methods (i), (ii), (iii) discussed in the rebuttal will be included for a quantitative comparison in transfer learning and the other applications and not just comparing the values of OTDD with different methods (fig 1 of the rebuttal) which is not informative; the order of magnitude does not tell anything on the discriminative power of a distance. 2) From the runtime plot in the rebuttal, It seems that an inner Sinkhorn is actually faster than OT-N for all dataset sizes. Could it be explained by the fact the dimension of MNIST is large making Bures too costly to compute ? Would you agree that for large d, Sinkhorn is better than OT-N and otherwise for large d ? ############ Overall this is a good paper; experiments were performed with great care including several datasets both in Vision and Text to illustrate how OTDD can quantify the learning "transferability" across datasets. My main concern is that while these results are promising, no baseline was provided to quantify the performance gain of OTDD. Ideally, comparisons with Gromov / Hierarchical OT discussed in the intro should have been done, however given that the main contributions of this paper are (1) use OT in the context of quantifying datasets proximity; (2) engineer a novel ground cost d_X to define the OT; I would at least provide some empirical comparisons to support these claims by: 1) Check the ground cost: replace the labels-labels ground cost (Bures-Wasserstein OT_N) by the Euclidean distance between the centroids of the conditional distributions (basically ignoring the variance term); this was discussed in L150-155 but discarded as a naive approach, still a naive baseline would have been a good sanity check to ensure that running Newton-Schultz to compute OT_N is worth it). Another sanity check could be to test d_UB (eq 8) instead of OT_N. And perhaps d_OT on a small subsampled dataset for the counter argument to validate the approximation. 2) Check OT: use an MMD distance with the proposed ground cost as a Kernel to see how whether the gain in performance is significantly explained by the geometric properties of OT. Given that the last experiments (fig 5 and 6) illustrate the same message, I invite the authors to dedicate this space to comparisons with the methods above. These methods would be significantly easier to compute. It would be nice to see whether the gain in performance -- if any -- is worth the burden of OT / OT_N. Other Questions =============== 3. Was d_X always taken as the Euclidean cost between features ? I guess in the context of images, it would be more appropriate to use d_X = OT to compare individual images (Using this version of Hierarchical OT would certainly be computationally problematic) but I'm actually wondering whether the geometric properties of OT would be better employed to compute the ground cost (+ MMD outside perhaps ?). 4. in the definition d_Z = d_X + d_OT; is there any reason for both d_X and d_OT would have a similar order of magnitude ? Could there be a scenario where the pair of distances should be weighted differently ? 5. I'm afraid I don't really see how the upper bound d_UB of Prop 4.1 is useful here ? OK d_OT_N is a lower bound (intuitively using the first two moments to compare the distributions provides a lower bound for OT) but given that the difference is not bounded I don't see how this makes the approximation legitimate; except by the quantity d_UB - d_OT_N which was not analyzed ? minor typos: L89: a inner-level L182: thisa L197: we briefly *show* how L241: fully *connected*


Review 3

Summary and Contributions: The paper introduces a new distanced based on optimal transport on both the samples and their labels. It first treats the labels as a distribution over the sample space and use the Wasserstein distance there to measure the labels. Then, it wraps the Wasserstein distance over the labels with the Euclidean distance over the samples in the sample space into the ground metric for the OT problem in the outer-layer. The distance satisfies all metric properties and is lower-and upper-bounded by its OT variants. Some computational tricks are included in the method to reduce its complexity and the results of several experiments reveal its applicability to different ML problems.

Strengths: -- soundness of the claims (theoretical grounding, empirical evaluation), The derivation of the method is sound so far (I will take a second look at the supplementary later). The empirical evaluation is not clear as for now, pending the response from the authors to my questions below. -- significance and novelty of the contribution, This is a novel work to the best of my knowledge. It is a good attempt in finding the statistical bond between the labels and the sample instead of adding them together as in previous work. The contributions of the theorems are valuable and their hardness is relatively low. -- relevance to the NeurIPS community. The paper address an important open problem in the ML community which is combining the feature space and the sample space. It can also attract attention to OT-based solutions which are increasingly popular.

Weaknesses: --Line 178 "Instead, we propose an alternative representation of the $alpha_y$ as Gaussian distributions". Isn't this assumption too strong? I can accept it for a well-embedded dataset but please help me understand why we can make such an assumption if it "(ii) does not involve training"? -- Line 233 The complexity O(n^2d + d^3) might be troublesome especially for high-dimensional features because of the d^3 term. n^2d might be OK because we can batch the computation in practice. If I miss it somehow, --Line 234 I might miss it somewhere but I cannot find the comparison with any baseline distances. -- Line 268, 284 What are the blue bands in Figure 5 and 6? I wouldn't call them "strongly correlated" if the p-value is 0.02 in 5 (a) or 0.01 in 6 (a) which barely exceeds the threshold.

Correctness: I haven't found any errors in the derivation of the method, although I think there are typos in Line 495 where Minkowski inequality is applied. Section 6, Experiments, is not clear enough to be evaluated as I pointed out below.

Clarity: The first five sections are clear. I got lost starting at Section 6 Experiments because I cannot clearly understand how OTDD is integrated into the learning iterations of the LeNet. Is it the loss? For classification? Is it supervised or unsupervised? After reading the first a few sections, I thought we could just compute the distance between two datasets but then it seems the datasets are embedded by LetNet-5? And then the last few layers are being fine-tuned? I got lost there. I also cannot see benefit (i) ~ (iv) of OTDD from Section 6. Please use 6.1 as an example to detail the use of OTDD.

Relation to Prior Work: Mostly, yes. -- JDOT by Courty et al. and its spin-off DeepJDOT by Bhushan Damodaran et al. are two important attempts in combining the sample space and the feature space. I think one of them should be cited in Related Work. Courty, Nicolas, Rémi Flamary, Amaury Habrard, and Alain Rakotomamonjy. "Joint distribution optimal transportation for domain adaptation." In Advances in Neural Information Processing Systems, pp. 3730-3739. 2017. Bhushan Damodaran, Bharath, Benjamin Kellenberger, Rémi Flamary, Devis Tuia, and Nicolas Courty. "Deepjdot: Deep joint distribution optimal transport for unsupervised domain adaptation." In Proceedings of the European Conference on Computer Vision (ECCV), pp. 447-463. 2018. -- What are the connections and differences between this work and Lee et al. in NeurIPS 2019 and an article on arXiv by Redko et al.? Lee, John, Max Dabagia, Eva Dyer, and Christopher Rozell. "Hierarchical optimal transport for multimodal distribution alignment." In Advances in Neural Information Processing Systems, pp. 13474-13484. 2019. Redko, Ivegen, Titouan Vayer, Rémi Flamary, and Nicolas Courty. "CO-Optimal Transport." arXiv preprint arXiv:2002.03731 (2020).

Reproducibility: No

Additional Feedback: Line 34 "..., (-are) highly depend(-ent) on..." might better serve the flow of the sentence. Line 182 "this(-a)" ------ After rebuttal I raised my rating by 1 point to acknowledge the rebuttal. The authors answered most of my questions. Still, my main concerns are the d^3 complexity, the baselines, and p-values as I mentioned in weakness. I hope the authors can add all the things as promised and put more focus on validating its practicality and impact through experiments on more datasets. MNIST is becoming more like a toy dataset these days.


Review 4

Summary and Contributions: This work proposes a geometric way of comparing datasets using optimal transport. This results in an OT problem which ground cost alleviate both the natural distance between the features of the dataset and a cost between the labels, the latter being defined as an OT problem itself between the conditional probability distributions of samples wrt the labels. Authors make the assumption that these inner transportation problems are transports between Gaussian distributions which can be computed more efficiently than computing all Wasserstein distances between general distributions. As a consequence of their approach this OT distance can be defined between two datasets even if the labels of the two datasets are unrelated, i.e. when there is no notion of distance between them. Authors argue that this distance is meaningful for comparing datasets and that transferring knowledge from a closer dataset wrt to their distance is more beneficial than transferring knowledge from a distant one. They show that empirically with transfer learning tasks with images and text datasets in a classification context.

Strengths: Overall I like the article, it is certainly useful for a wide range of tasks, and formalizes a natural, but quite challenging problem. Although the main idea is simple I find clever to alleviate the natural distance on the features in order to define a meaningful distance on the labels. Moreover in practice the correlation between "accuracy" and "closest dataset" indicates that this notion of distance might be useful. The paper is also well written and enjoyable to follow, I thank the authors for that. - Pros: (1) Well written paper, easy to follow. (2) Tackle an interesting problem with an elegant method. I think this is quite novel. (3) Experimental section seems sounded and illustrates well the usefulness of the approach.

Weaknesses: - Cons: (1) The method could be more justified, especially wrt relative importance of the feature and class terms and the use of OT for the distance for the labels. (2) The gaussian assumption could be more justified, especially since d_OT is not used anywhere in the paper it is hard to tell if it is a reasonable assumption and Proposition 4.1. is not particularity enlightening for that. In the following I will call "outer" Wasserstein the Wasserstein with ground cost d(x,x')+W(alpha_y,alpha_y') and the "inner" Wasserstein distances the terms W(alpha_y,alpha_y'). --- About the method --- (1) I think the relative importance of the terms d(x,x') and W(alpha_y,alpha_y') should be highlighted. How do you properly scale between the two costs for the outer Wasserstein ? Maybe it would be useful to trade-off between the feature cost d(x,x') and the Wasserstein W(alpha_y,alpha_y') cost with a parameter since they might be several order of magnitudes different which could induces flaws in the outer coupling. Did you face this problem in practice and did you handle it somehow ? (2) Related to the previous remark, it is not clear if the term d(x,x') has an important impact, since in W(alpha_y,alpha_y') there is also the feature information, with the additional class information. Is d(x,x') important or just realigning the conditional distributions is sufficient ? For example in Figure 1 and Figure 3 how does it behave without d(x,x') ? (3) I think also Figure 1 should be more detailed, it is not so clear here that this is the expected behaviour. (4) Since the inner Wasserstein distances drive the computational complexity why not using for example something like MMD here or a divergence ? The outer OT already finds the correspondences between the samples. It is not straightforward to me why OT is really mandatory for the inner costs, maybe it should be valuable to motivate a little bit further the use of OT. --- Theoretical results --- (1) I am not really convinced by the usefulness of Proposition 4.1. I do not see the interest of the upper bound here, since we could have taken any bound with a suboptimal coupling like the product coupling. This result is not particularity enlightening and only illustrates the fact that OT between Gaussian is a lower bound of the true OT. The Gaussian assumption seems more a heuristic and convenient choice for computational purposes. It implies that we restrict the transport between the conditional probabilities to be linear (in term of Monge map) [2]. Although this is reasonable since having an O(n5) complexity is prohibitive, Proposition 4.1. in this form does not appear to be a strong argument for this assumption. I think it would be valuable to test this assumption for toy datasets by comparing the behavior of d_OT compared to d_OT_N. (2) The distance property could be more explained by properly defining the space Z which is, I think, defined as Z=X * P(X|Y). I assume that having a "true metric" on datasets is quite vague in this case as we can not define and distance on the labels and rely on a OT distance for the joint distributions P(X*Y) as in [1]. (3) Although the assumption that transferring knowledge from closest dataset leads to better improvements than transferring from distant one seems reasonable do authors have some theoretical hints for this ? Although I understand that it is not in the scope of this paper I think discussing domain adaptation using this distance could be beneficial, as it can handle the case where there is no notion of distance between the labels, contrary to [1]. For example I am very curious to see if the scenarii where the adaptation is known to be difficult is correlated with the fact that the datasets are far from each others. [1] Joint distribution optimal transportation for domain adaptation, Nicolas Courty, Rémi Flamary, Amaury Habrard and Alain Rakotomamonjy. NeurIPS 2017. [2] Concentration bounds for linear Monge mapping estimation and optimal transport domain adaptation, Rémi Flamary, Karim Lounici, André Ferrari. 2019.

Correctness: Overall I think the claims are correct, I checked the proofs in the supplementary and I found no mistakes. The empirical methodology is solid, I like the experiments.

Clarity: The paper is well written and easy to follow.

Relation to Prior Work: I think this work is quite novel and it is hard to relate with other works. I am not aware of any other works on transferring knowledge based on a notion of distance between dataset but I am not an expert in transfert learning so I might be wrong.

Reproducibility: No

Additional Feedback: --- Small comments --- I think [3] is a related work since it also defines a distance between datasets. Also [1] should be cited in the OTDA part as it handles the case where there is a cost between the labels and relies on a distance between the joint distributions P(X*Y). References: [1] Joint distribution optimal transportation for domain adaptation, Nicolas Courty, Rémi Flamary, Amaury Habrard and Alain Rakotomamonjy. NeurIPS 2017. [2] Concentration bounds for linear Monge mapping estimation and optimal transport domain adaptation, Rémi Flamary, Karim Lounici, André Ferrari. 2019. [3] CO-Optimal Transport, Ievgen Redko, Titouan Vayer, Rémi Flamary, Nicolas Courty. 2020. ############### AFTER REBUTTAL ############### After the rebuttal I decrease my score of 1. Although I am confident that this is a good work the concerns raised by the other reviewers prevent me for giving this article a clear accept. I think authors should give the simple baselines and sanity checks in their main experiments. It is not very clear in this form that the OT approach really worth it compared to more "naive" approaches such as MMD, or without the term d(x,x'). I am fairly confident that this work is interesting for the ML community and I thank the authors for the efforts put in the writing since the article is very enjoyable to follow. I believe that if the baselines discussed in the rebuttal are included this will make the article much stronger. ############### AFTER REBUTTAL 2 ############### After some hesitations I finally decided to keep my initial score of 7. The only thing that prevented me for giving a 7 is the additional baselines discussed in the rebuttal. However after some thoughts I am fairly confident that the authors would provide them and that it would not hurt the quality of the paper. Authors tackle this point in the rebuttal and say that when considering the baselines there is a drop in the performances. Moreover when removing the features, authors say that the performance does not change: it is not a weakness since this case can be easily tackled using a trade-off parameter between the feature term and the inner Wasserstein (the method is quite generic). The MMD approach instead of inner Wasserstein distances could be competitive and I think this article paves the path for such interesting studies. I believe that the article has enough good contributions and IF the authors do add the baselines described in the rebuttal it would be a shame if it is not accepted. I am more inclined to give it a chance by keeping my initial score of 7.

[Author Response · NeurIPS 2020]



Figure 1: Comparison of methods to compute inner OT distance $d_{\mathcal{Y}}$ in OTDD between independent MNIST samples.

**General Comments**: We thank all reviewers for stupendous feedback, which has greatly improved the paper.

• **Gaussian Approximation** (R3/R5). Reviewers justifiably worry that this approximation could sometimes be too
coarse. As R3 points out, in cases where the data is first embedded with some complex non-linear mapping (neural
net or otherwise), there's empirical evidence that the first two moments capture enough relevant information for
classification (Seddik et al, 2020). Note that this is the case for our text classification experiments. But we would argue
that in general, despite the coarseness of this approximation, OTDD is objectively capturing relevant information (as
shown by our experimental results), so any refinement over this approximation can only further improve the quality
of the distance. For small datasets, computing the exact $d_{\mathrm{OT}}$ is feasible and—as predicted by Thm 5.1—sometimes
even faster that $d_{\mathrm{OT}-\mathcal{N}}$, but for very large datasets ($n \gg d$) the latter will often be the only viable option (Fig 1).

• **Bounds** (R2/R5). We agree that the upper bound in Prop 4.1 is much less informative that the lower one (as is often
the case in OT), but still, as R2 points out, having *any* upper bound allows one to e.g. bound the UB-LB band around
the exact OTDD, as shown in Fig 1.

• **Baselines** (R2/R3). The lack of other distances that allow for datasets with non-identical label sets limits possible
baselines except in simple settings. We will include those suggested by R2 plus some additional OTDD variants:
(i) $d_{\mathcal{Z}} = d_{\mathcal{X}}$ (ignore labels), (ii) $d_{\mathcal{Z}} = d_{\mathcal{Y}}$ (ignore features), (iii) $d_{\mathcal{Y}} = \|\mu_y - \mu'_y\|$, (iv) $d_{\mathcal{Y}} = \mathcal{L}(y, y')$ (i.e., JDOT,
possible if some pairs). Limited space precludes detailed results here, but we have now verified that these ablations
weaken correlation with transferability on *NIST: e.g., $\rho$ drops by 5-10% for (i)/(ii), and (i) loses significance (at
$\alpha = .05$), though interestingly (ii) doesn't (c.f. R5's point). We will include these results in the final version.

• **Relation to JDOT/DeepJDOT**. We thank R3&R5 for raising this. It was an omission on our part not to discuss
these very relevant works, but the revised version does. Summary: main difference with OTDD is that those rely on a
classification loss to measure label similarity, which (i) requires same label sets across domains, (ii) depending on $\mathcal{L}$
might not yield a true metric. We will nevertheless compare against these where possible (i.e., MNIST↔USPS).

• **Scaling/weighting of $d_{\mathcal{X}}$ vs $d_{\mathcal{Y}}$** (R2/R5). Given the interpretation of $d_{\mathcal{Y}}(y, y')^p = \mathrm{W}_p^p(\alpha_y, \alpha_{y'})$ as an expectation of
$d_{\mathcal{X}}^p$, the two terms in $d_{\mathcal{Z}}$ are by construction of the same order, and—we argue—should be equally weighted barring
additional information. Yet, as mentioned by R2/R5, it might be interesting to weight these two terms differently in
specific settings - we appreciate the suggestion and have included this option in our codebase.

**Reviewer 1**: Without assumptions on labels, cond. distrib is arguably the *only* info about $y$ we have. We *do* use entropic
OT for outer problem (cf. Thm 5.1) & can also be used for inner one, but Gaussian approx. is faster for large $n$ (Fig 1).

**Reviewer 2**: We appreciate the many suggested baselines/ablations, some of which we have now implemented (Fig 1)

• Gromov / Hierarchical OT cannot—in their usual form—incorporate discrete and distinct label sets. We haven't
observed much difference between various 'feature-only' metrics, but will include them nevertheless.

• The suggested ablation based on MMD is intriguing - we will try it and include in the revised version.

• $d_{\mathcal{X}}$ was always Euclidean here, but yes, it could be taken as OT distance! (additional cost/quality trade-off)

**Reviewer 3**: We hope the two points below clarify the doubts regarding experimental evaluation.

• "how is OTDD integrated into learning " $\rightarrow$ it is not. OTDD is computed directly on the datasets $\mathrm{D}_s, \mathrm{D}_t$. *Separately*,
we pretrain net on $\mathrm{D}_s$ and fine-tune on $\mathrm{D}_t$, w/ usual supervised training. Goal: show OTDD *predicts* transfer success.

• The bands in Figs 5-8 are 95% conf. intervals via bootstrap on 10 repetitions. "*wouldn't call them strongly correlated*":
note that strength of correlation refers to the slope $\rho$. As for significance, a p-value of $0.02$ might be considered
'mildly' significant, but the other three settings yield p-values $\in (10^{-5}, 10^{-2})$, significant by any standard.

• Complexity. Indeed, for $d \gg n$, $d^3$ dominates, which suggests (exact) $d_{\mathrm{OT}}$ is actually preferable (see Fig 1).

**Reviewer 5**: We are grateful for the detailed+acute observations/suggestions. We've adopted inner/outer terminology.

• As you suggest, following proof technique of JDOT paper we are able to show a similar adaptation bound - will add

[1] Seddik et al., *Random matrix theory Proves that Deep Learning representations of GAN-data behave as gaussian mixtures*, 2020.

[Meta-Review · NeurIPS 2020]

The paper proposes an interesting approach and investigates the question of dataset similarity that in an important problem for transfer learning. Still it was clearly borderline in the initial review round. The feedback from the authors was appreciated and answer some of the concerns from the reviewers who reached a weak accept consensus. Note that the final version must take into account the comments from the reviewers especially the baseline comparison discussed in the feedback. More precisely the authors are expected to do the following for the final version: - Add the baselines discussed in the rebuttal in the experiments (ignoring features and labels, JDOT) and the ones discussed by R2 (distance between means) since it is an approximation of Bures and would illustrate the importance of the second order moments. - Add the discussion about JDOT and in addition to the baselines.